# Role of paleogeography on large-scale circulation during the early Eocene

Fanni Dóra Kelemen<sup>1</sup>, Richard Lohmann<sup>1</sup>, Jiang Zhu<sup>2</sup>, and Bodo Ahrens<sup>1</sup>

<sup>1</sup>Institute for Atmospheric and Environmental Sciences, Goethe University Frankfurt, Frankfurt am Main, Germany

<sup>2</sup>NSF National Center for Atmospheric Research, Boulder, Colorado, USA

Correspondence: Fanni Dóra Kelemen (kelemen@iau.uni-frankfurt.de)

Abstract. The configuration of continents and oceans has a major influence on Earth's climate by shaping large-scale atmospheric circulation patterns. In this study, we investigate the effect of early Eocene paleogeography, specifically from the Ypresian stage, on extratropical eddies. We analyse the influence of the epicontinental West Siberian Sea as well as the impact of the absence of the Antarctic Circumpolar Current on mid-latitude cyclones and blocking events. Previous work from the Deep-Time Model Intercomparison Project (DeepMIP) has shown that during the early Eocene, heat transport through cyclonic systems was more intense than under modern conditions at the northern mid-latitudes, and less intense at the southern mid-latitudes. We analyse cyclone tracks and blocking systems of the early Eocene in an atmosphere-only CESM1.2 simulation, continuing the DeepMIP 1xCO<sub>2</sub> experiment. Sea surface temperatures from the DeepMIP experiment are used as boundary conditions. The simulation output is six-hourly, which enables direct cyclone tracking in the pressure field. In parallel, a decrease in heat transport of stationary eddies at the northern mid-latitudes in DeepMIP data, motivates the analysis of blocking climatology based on the 500 hPa geopotential height field. Our results show that, through air–sea interactions, paleogeographic features of the early Eocene enhance cyclonic activity at northern mid-latitudes while reducing it at southern mid-latitudes compared to modern conditions.

keywords: paleogeography, Early Eocene Climatic Optimum, mid-latitude cyclone, blocking, West Siberian Sea

#### 15 1 Introduction

The arrangement of continents and oceans has an important effect on climate, from regional to global scales. Throughout Earth's history, paleogeography has shaped key aspects of the climate system, including atmospheric and oceanic heat transport, the hydrological and the carbon cycle. The continental configuration strongly influences ocean currents, for example the development of the Antarctic Circumpolar Current (ACC) was enabled by the opening of ocean passages (Scher et al., 2015). Moreover, it has been shown that C-shaped distribution of land at the tropics or subtropics favours monsoon development (Mei et al., 2025). Also, tectonic changes have the capacity to trigger ice ages, through their influence on the carbon cycle. When the majority of continents are located at the tropics and experience tropical precipitation, silicate weathering can effectively remove  $CO_2$  from the atmosphere inducing a cooling and ice accumulation (Ramstein, 2011). In paleoclimate research, attributing different forcing mechanisms is essential for understanding climate dynamics and for making conclusions transferable

45

Figure 1. Early Eocene paleogeography from Herold et al. (2014).

to also future scenarios. Moreover, studying past warm climates are of high relevance because of their information on how our climate system worked under high  $CO_2$  concentrations.

Here, we consider the role of the early Eocene paleogeography on large-scale circulation patterns. The time period we focus on is the early Eocene Climatic Optimum (EECO,  $\sim 53-51$  Ma), which is characterised by exceptional warmth (approximately  $10-16^{\circ}$ C warmer than preindustrial climate) (Inglis et al., 2020), low meridional temperature gradients (Evans et al., 2018) and high  $CO_2$  concentrations (between 1,170 and 2,490 ppm) (Anagnostou et al., 2020). There has been coordinated effort amongst modelling and proxy groups, under the initiative known as the Deep-Time Model Intercomparison Project (DeepMIP), to increase our understanding of this period in Earth climate history (Lunt et al., 2021). In our study, we consider the paleogeography used in the first phase of DeepMIP, from the early Eocene, the Ypresian stage representing the Earth between  $\sim 55$  to  $\sim 50$  Ma (see Fig. 1) (Lunt et al., 2017; Herold et al., 2014).

The most notable features of this early Eocene paleogeography are the wider Pacific Basin and thus narrower Atlantic Ocean, the tropical location of the Indian subcontinent and the lack of the Himalayas. Moreover, an important characteristic in the Northern Hemisphere is the connection between the Arctic and the Tethys Ocean through the shallow West-Siberian Sea (Shellito et al., 2009; Wang, 2004). At this time in the past, the Bering Strait was closed and there were no currents connecting the Arctic to the Pacific basin, thus oceanic heat transport from the tropics to the Arctic flow through the West-Siberian Sea (Akhmetiev et al., 2012). This is underlined by proxy records showing that the mean annual sea surface temperature in the Arctic Ocean and the West-Siberian Sea was very similar and at least 20°C (Frieling et al., 2014). In the Southern Hemisphere, the location of the Antarctic continent was very similar to modern conditions, but both Australia and South America were located more to the south, causing the Drake Passage and the Tasman Gateway to be narrow and shallow, i.e. oceanographically closed (negligible water, heat or mass transport through it) (Herold et al., 2014) and thus inhibiting the development of the ACC.

Previous analysis of the DeepMIP ensemble showed that even though the total meridional heat transport (MHT) does not change much due to the Eocene boundary conditions in the model simulations, there are changes in the atmospheric heat transport (AHT), and the ocean heat transport (OHT) shifts towards the South Pole (Fig. 2a) (Kelemen et al., 2023). This is

Figure 2. Annual meridional heat transport (MHT) and its partitions from the DeepMIP based CESM1.2 pre-industrial (solid) and  $1xCO_2$  (dashed) simulations. (a) MHT divided into atmospheric (AHT) and oceanic (OHT) heat transport, (b) AHT divided into meridional overturning circulation (MOC), stationary eddies (SE) and transient eddies (TE) for further details on the calculations see Kelemen et al. (2023).

likely related to the fact that, in most DeepMIP models, deep-water formation occurs in the Southern Ocean (Zhang et al., 2022). Thus, due to the change in deep-water formation, respective to modern days, the hemispheric distribution of OHT changes as well. Kelemen et al. (2023) also showed that the partitions of AHT, namely the heat transport through different physical processes such as the atmospheric meridional overturning circulation (MOC), the stationary (SE) and the transient eddies (TE), also change due to the paleo boundary conditions. Most notable is the increase in TE heat transport at the northern mid-latitudes, and the compensating decrease in SE heat transport. Meanwhile, over the southern mid-latitudes, a decrease in TE heat transport is identified in the data (Fig. 2b), which is not compensated by other processes, thus mainly responsible for the decrease in AHT over the Southern Hemisphere. The decrease in AHT is also linked to the aforementioned increase in OHT over the Southern Hemisphere, consistent with the Bjerknes compensation mechanism.

In this study, we further investigate the changes seen in the heat transport analysis (Kelemen et al., 2023), and analyze how the early Eocene paleogeography influences large-scale circulation patterns, especially at the mid-latitudes where AHT and its differences are maximal. We are focusing primarily on mid-latitude circulations patters, cyclones and blockings. For the analysis we use data from CESM1.2 model simulations, which are the extensions of the DeepMIP simulations (Zhu et al., 2019; Lunt et al., 2021).

Our goals in this paper are:

65

- 1. To understand the early Eocene global circulation and climate. The better understanding of large-scale circulation patterns helps the interpretation of proxy records and the separation between signals related to paleogeography and  $CO_2$  concentration.
- 2. To demonstrate how AHT change materialises through dynamical processes, and to show how surface interactions govern large-scale atmospheric circulation patterns.

This paper is structured as follows. In section 2, we introduce the climate model simulations and the identifying algorithms for cyclones and blockings. In section 3, we provide an overview of our results considering the Northern and Southern Hemisphere. In section 4, we discuss our findings and the wider context of the dynamic changes and finish with a summary and our conclusions.

#### 2 Data and Methods

#### 2.1 Climate Model Data

The study uses the Community Earth System Model version 1.2 (CESM) in an atmosphere-only configuration, where the atmosphere and land modules are active, while the ocean is taken into consideration through prescribed monthly Sea Surface Temperature (SST) and sea-ice extent. For other details on the model and boundary conditions, see Zhu et al. (2019) and Lunt et al. (2021). For our analysis, we utilised two simulations; a pre-industrial control simulation and a  $1xCO_2$  EECO experiment, where all boundary conditions are changed to represent the EECO conditions, except for the  $CO_2$  concentration, which is kept at the pre-industrial level. The differences between the modern and EECO simulation are the changes in paleogeography, the lack of ice sheets, vegetation, aerosols and rivers. Thus, comparison of these simulations reveals the effect of these boundary conditions, with paleogeography hypothesized to make the largest contribution.

The simulations were extended from the corresponding coupled simulations in the DeepMIP ensemble (Lunt et al., 2021). The motivation for the extension is the need for at least 6 hourly resolution to enable direct cyclone tracking, which is not possible in the original DeepMIP dataset's monthly data. The initial conditions were taken from the final state of the DeepMIP simulations, and monthly SST as well as sea ice fraction values were calculated from the last 100 years of the simulations. Our simulations were integrated for 30 model years to represent the climatology of the climate state.

## 2.2 Cyclone Tracking

To identify cyclones in the simulations, we adapted the cyclone tracking algorithm of Kelemen et al. (2015) for the global field. The algorithm identifies cyclones as mean sea level pressure minima and connects the pressure centres in time by a nearest neighbour approach.

For the climatological analysis we do not consider the trajectories one-by-one, but calculate a track density field from them. The track density field represents the average number of cyclone centre passing through each model grid point per year or per season. The field is calculated through the cyclone centres, thus the area where cyclones affect the climate is larger and

surrounds the high track density storm tracks. To focus on the significant transient eddies, our analysis includes only cyclones that persist for more than one day, travel at least 1000 km, and exhibit a pressure minimum at least 20 hPa below the global mean pressure.

## 2.3 Blocking calculation

The applied hybrid blocking index evaluates the geopotential height at 500 hPa. This hybrid index combines the gradient approach by Davini et al. (2012) with an anomaly approach based on Barriopedro et al. (2010) which we modified as described in Lohmann et al. (2024): in case of blocking the geopotential height has to exceed the climatological mean by one standard deviation. The mean value and standard deviation were calculated for each calendar day and grid cell. A 91-day window centred around the day of interest was applied for the calculation of the mean and standard deviation to smooth the yearly cycle. The climatological mean of geopotential height refers to a 31-year running window to consider changes in the mean geopotential height related to changes in global mean temperature.

In the first step, both components of the hybrid index were calculated independently. To combine both, blocked areas were checked for a joint area of at least  $1.5 \cdot 10^5$  km<sup>2</sup>. If this threshold was exceeded, the full area detected by the anomaly approach was counted as instantaneous block. Next, a spatio-temporal filtering was applied. The instantaneous block was finally counted as block if the block persists for at least five days, covers at least  $15^{\circ}$  in longitude and  $1.5 \cdot 10^6$  km<sup>2</sup> in space. The blocking climatologies were calculated with the 2D-Blocking Plugin (Richling, 2020) from the Free Evaluation System Framework (Freva) (Kadow et al., 2021). The geopotential height fields were remapped to a  $2.5^{\circ}$ x $2.5^{\circ}$  grid, since this spatial resolution is the default setting of the 2D-Blocking Plugin (Richling, 2020).

## 3 Results

The analysis of the cyclone tracks shows a change in spatial pattern in both Hemispheres between the pre-industrial and the  $1xCO_2$  simulations (Figure 3). Under pre-industrial conditions, mid-latitude cyclones in the Northern Hemisphere are concentrated around two distinct areas over the Pacific and Atlantic Oceans. On the other hand, Southern Hemisphere cyclone tracks distribute around Antarctica in a O shape. However, under early Eocene conditions, the distinction between the two sets of Northern Hemispheric cyclone paths becomes less pronounced. Also, there is an increase in cyclone tracks passing over land areas, especially over Europe. In contrast, in the Southern Hemisphere the cyclone tracks show a more dispersed distribution, with several small density centres around Antarctica, due to the more fragmented Southern Ocean basin. These changes are also reflected in the number of cyclones (Figure 4), with an average increase of 36% in the Northern Hemisphere and a decrease of 32% in the Southern Hemisphere. The changes are consistent through all seasons, but most prominent during each hemisphere's winter and spring season. These findings are in line with the increased transient eddy heat transport in the northern mid-latitudes, respectively (Figure 2).

Early Eocene blocking frequencies in the northern mid-latitudes show a general decrease and a shift from modern Europe to the east to the West-Siberian Sea (Figure 5a and b). This is also in line with the heat transport analysis, showing a decrease in the

Figure 3. Global annual mean track density under pre-industrial (a,c) and early Eocene  $(1xCO_2)$  (b,d) conditions.

stationary eddy heat transport at the northern mid-latitudes (see Figure 2). On the other hand, the heat transport analysis does not show large changes in the stationary eddy heat transport at the southern mid-latitudes, which is reflected in the blocking

Figure 4. Seasonal distribution of cyclone numbers in the pre-industrial and early Eocene  $(1xCO_2)$  simulations over the (a) Northern Hemisphere (b) Southern Hemisphere.

frequencies' similar intensities. Nevertheless, in the early Eocene the southern blockings have a more dispersed distribution around Antarctica than in the pre-industrial climate (Figure 5c and d).

Mid-latitude cyclones play a key role in the hydroclimate as they transport moisture and are associated with a large portion of extratropical precipitation. For qualitative measure we plotted boreal winter (December, January and February) and southern winter (June, July and August) cyclone track densities along the respective seasonal precipitation fields (Figure 6 and 7). In the Northern Hemisphere, cyclones are more prominent during winter and they precipitate along the Tethys Ocean and the West-Siberian Sea, bringing moisture to an area which is dryer under modern conditions. Moreover, during the early Eocene the west coast of North America is experiencing high precipitations probably due to the high number of tracks and orographic lifting. In the Southern Hemisphere there are more cyclones during the southern winter (see Figure 4) and they bring precipitation to the western coast of Australia and South America (Figure 7). Both of these regions are located souther in the early Eocene topography than in modern times, thus more in the paths of the mid-latitude cyclones.

### 4 Discussion and Conclusions

- The meridional heat transport analysis identified three main characteristics (Figure 2) when comparing the pre-industrial and  $1xCO_2$  Eocene simulations:
  - increase of transient eddy heat transport at the northern mid-latitudes
  - decrease of stationary eddy heat transport at northern mid-latitudes
  - decrease of transient eddy (and atmospheric) heat transport at southern mid-latitudes
- These characteristics are due to the paleo boundary conditions, most notably the different configuration of continents and their orography. The cyclone and blocking analysis shows that the increase of transient eddy transport over the northern mid-

Figure 5. Annual mean blocking frequency under pre-industrial (a,c) and early Eocene  $(1xCO_2)$  (b,d) conditions.

latitudes is due to the increase in cyclone numbers, the decrease of stationary eddy transport is due to less frequent blockings at the northern mid-latitudes. Moreover, the southern mid-latitude transient eddy heat transport decrease is connected to the decrease of cyclone numbers. This change is not compensated by other processes, thus effects the AHT as well.

Figure 6. Boreal winter (DJF) track density (left column) and mean precipitation (right column) in the (a,b) pre-industrial and (c,d) early Eocene ( $1xCO_2$ ) simulations.

The changes in paleogeography influence the air-sea interactions in the climate system. In the Northern Hemisphere, the Tethys Ocean and the West Siberian Sea served as additional sources of heat and moisture. This effect was especially pronounced during winter, when the thermal contrast between the epicontinental West Siberian Sea and the neighbouring land areas of Europe and Asia was highest. Comparing the latent and sensible heat fluxes over the Eurasian region highlights the role of the West-Siberian Sea in transporting heat and moisture to the atmosphere (Figure 8). This configuration made it possible for baroclinic instabilities to grow which we saw as an increase in cyclone numbers.

In the Southern Hemisphere, the most significant changes in paleogeography compared to modern are the closed Drake Passage and closed Tasman Gateway. These two narrow and shallow straights prevented the development of the wind driven circumpolar ocean current around Antarctica (ACC). The surface wind field is indeed weaker in the  $1xCO_2$  Eocene simulation than under modern circumstances (Figure 9). The ACC is dynamically connected to southern hemispheric extratropical cyclones, and its absence in the  $1xCO_2$  Eocene simulation is in line with the fewer cyclone number (Figure 4). The ACC is crucial in maintaining the strong oceanic temperature gradient between cold polar and warmer subtropical waters, this increases baroclinic instability in the region. It is shown that in modern climate the southern hemispheric jet stream is stronger than its northern counterpart, and that the Southern Hemisphere is stormier (Shaw et al., 2022). In the early Eocene conditions we observe the opposite signal, where the Northern Hemisphere is stormier.

Figure 7. Southern winter (JJA) track density (left column) and mean precipitation (right column) in the (a,b) pre-industrial and (c,d) early Eocene ( $1xCO_2$ ) simulations.

We conclude that through air-sea interactions the paleogeography influences atmospheric large-scale circulation patterns. Under early Eocene circumstances, this influence meant increased mid-latitude baroclinity over the Northern, but less over the Southern Hemisphere. In the Northern Hemisphere, this resulted in increased heat transport by large eddies, nevertheless this process seems to be compensated by less heat transport via blocking systems. On the other hand, in the Southern Hemisphere the decreased baroclinity and thus less heat transport via extratropical cyclones is not compensated by other processes in the atmosphere. The changes induced by paleogeopgraphy have a net effect on AHT, which is in turn compensated by the OHT. The increase in southward OHT is connected to the southern deep-water formation under early Eocene paleo boundary conditions (Zhang et al., 2022; Kelemen et al., 2023). From the energetic perspective, the change in AHT needs to be compensated by the ocean to fulfil the top of the atmosphere energetic constraint on MHT. Thus, the ocean needs to increase its heat transport over the Southern Hemisphere, which favours the development of a Southern Hemisphere-driven overturning circulation (Zhang et al., 2022).

This study shows that the early Eocene paleogeography affects atmospheric large scale processes and ocean circulation both in a direct and indirect way. For the ocean, the direct effects consist of strait geometry, and the indirect effects include the heat transport processes' influence on deep-water formation. Our results support the hypothesis that under early Eocene boundary conditions the Southern Ocean deep-water formation is more likely to fulfil the energetic constraints of the climate system.

185

190

Figure 8. Latent heat and sensible heat fluxes during boreal winter over the Northern Hemisphere in the (a) pre-industrial and (b) early Eocene  $(1xCO_2)$  simulations.

In this paper, we investigated the role of early Eocene orography on large scale circulation patterns. We compared atmosphere only CESM model simulations describing pre-industrial conditions and modern day orography versus  $1xCO_2$  simulations containing early Eocene boundary conditions including paleogeography, the lack of ice sheets, different vegetation and rivers. For ocean boundary conditions, we used the SST values form DeepMIP CESM simulations. Our research was motivated by a meridional heat transport analysis, which showed changes in the heat transport of transient eddies (cyclones) and stationary eddies (blockings) at the mid-latitudes. We found that the increase in transient eddy transport in the northern mid-latitudes is connected to the increase in cyclone numbers. In the early Eocene continental configuration, the presence of the warm and shallow epicontinental West-Siberian Sea enhances air-sea interactions and increases the baroclinic instability in the region. The change in cyclone numbers and paths also resulted in more moisture transport into the Asian continental interior. On the other hand, we found that the increased heat transport through cyclones is compensated by a decreased fraction of blocking under early Eocene conditions. Over the Southern Hemisphere, we found a decrease in mid-latitude cyclone numbers, which is connected to the decrease in temperature gradient between polar and subpolar surface water due to the lack of the ACC. We hypothesise that the decrease in southern AHT infers the increase in OHT which is achieved by the Southern Hemispheredriven overturning circulation in the ocean. We conclude that the pealoegeography has an important role in forming the large scale circulation through air-sea-land interactions, both in the atmosphere and in the ocean. The next step of this research is the investigation of early Eocene cyclones in a high  $CO_2$  simulation, which is representing the paleo conditions more accurately.

205

Figure 9. Horizontal wind at 10 m during southern winter (JJA) over the Southern Hemisphere in the (a) pre-industrial and (b) early Eocene  $(1xCO_2)$  simulations.

Code and data availability. The DeepMIP CESM1.2 simulations are available by following the instructions at https://www.deepmip.org/dataeocene/ and the restart files at Zhu et al. (2024). The atmosphere-only simulation data used for this study is available at Kelemen (2025)

Author contributions. Conceptualization and Investigation: FDK, Data curation: FDK, JZ, Formal analysis: FDK, RL, Funding acquisition: BA. All authors contributed to the writing, reviewing and editing of the manuscript.

Competing interests. The authors declare no competing interests.

Acknowledgements. This research was funded through the VeWA consortium (Past Warm Periods as Natural Analogues of our high-CO2 Climate Future) by the LOEWE programme of the Hessen Ministry of Higher Education, Research and the Arts, Germany. FDK and BA acknowledges support from Hessen Ministry of Higher Education, Research and the Arts (Hessisches Ministerium für Wissenschaft und Kunst, Grant 67). This work used resources of the Deutsches Klimarechenzentrum (DKRZ) granted by its Scientific Steering Committee (WLA) under project ID1346. The CESM project is supported primarily by the National Science Foundation (NSF). This material is based

upon work supported by the National Center for Atmospheric Research (NCAR), which is a major facility sponsored by the NSF under Cooperative Agreement No. 1852977.

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
