# Peer review of "Role of paleogeography on large-scale circulation during the early Eocene"

_EGUsphere, 2025_

## Author Comment (AC1)

**RC1**: 'Comment on egusphere-2025-4923', Anonymous Referee #1

This manuscript is well-organized contribution to our understanding of how early Eocene paleogeography shaped mid-latitude circulation, cyclone activity and blocking. The modeling setup is appropriate, and the comparison between pre-industrial (PI) and early Eocene conditions is clearly motivated. The main conclusions are convincing and broadly consistent with earlier DeepMIP work. That said, some sections are quite dense, a few claims could use clearer supporting evidence, and there are places where the writing becomes a bit repetitive. With some tightening and clarification.. the paper will be even stronger.

We thank the Reviewer for their comments and suggestions, we considered them one by one and we feel that the manuscript indeed became clearer and easier to follow. Below, we list our answers and the changes in the manuscript according to each comment.

Major Comments

1. Lines 5–10: The abstract is informative but a bit crowded. You could shorten the background on DeepMIP so the focus stays on what this study finds.

   We have shortened the sentence and made it a bit more general.

   "Previous work from the Deep-Time Model Intercomparison Project (DeepMIP) has shown changes in atmospheric eddy heat transport under early Eocene boundary conditions."

2. Line 10: The phrase "In parallel, a decrease in heat transport…" reads abruptly. Maybe say something like: "Motivating our blocking analysis, previous DeepMIP results showed…"

   We have shortened and rephrased this part of the abstract.

3. The core message.. more cyclones in the Northern Hemisphere and fewer in the Southern Hemisphere.. Is clear and well placed. The paleogeographic background (Lines 15–30) is good, but dense. A single summary line linking the geological setup to atmospheric circulation would help.

   We have changed the paleogeographic background paragraph (see comment 5). In the abstract now we included the line: "Our results show that, through air–sea interactions the paleogeographic features of the early Eocene produce a more balanced heat transport between the hemispheres and atmospheric processes compared to pre-industrial conditions."

4. Line 21: The "C-shaped distribution of landmass" might confuse some readers. Adding a short explanation or pointing to a figure would help.

   We included a line explaining the C-shaped land configuration.

"The continental configuration also influences atmospheric circulation. For example, continents with a C-shape land distribution at the tropics, two continental landmasses at the two sides of the Equator connected with a bridging land, favours monsoon development (Mei et al., 2025)."

5. Lines 24–31: This section jumps quickly between monsoons, carbon cycle changes, and ice ages. Tightening it around "how paleogeography shapes climate circulation" would improve the flow.

We agree, that it needed a clearer flow, we changed it to:

"The arrangement of continents and oceans has an important effect on climate, from regional to global scales. Throughout Earth's history, paleogeography has shaped key aspects of the climate system, including oceanic and atmospheric circulation, and the carbon cycle. An example in the ocean is the Antarctic Circumpolar Current (ACC), which could only develop after ocean passages opened 30 million years ago (Scher et al., 2015). The continental configuration also influences atmospheric circulation. It has been shown (Agustsdottir et al., 1999) that through the geological time scale, paleogeography plays the most important role in influencing winter storm distribution. Also, monsoonal circulation is shown to be dependent on the continental configuration. For example, continents with a C-shape land distribution at the tropics, two continental landmasses at the two sides of the Equator connected with a bridging land, favour monsoon development (Mei et al., 2025). In another aspect, when the majority of continents are located at the tropics and experience tropical precipitation, silicate weathering can effectively remove $CO_2$ from the atmosphere inducing a cooling and ice accumulation (Ramstein, 2011). Thus, tectonic changes have the capacity to trigger ice ages, through their influence on the carbon cycle."

6. Lines 35–45: The EECO description is well supported, but the number of proxies cited slows the narrative. You could trim a bit here.

We agree that citations can affect the readability of the text. In the mentioned paragraph there is 1 and in one instance 2 references per sentence, referring to facts, which we do not wish to delete, because they help to establish the scientific basis of our study.

7. Lines 45–55: When referring to Kelemen et al. (2023), it would help to briefly restate what changed in the heat transport components (e.g., stronger transient eddy transport in NH).

The section explaining the results from Kelemen et al. (2023) reads now as follows:

"Kelemen et al. (2023) also showed that the partitions of AHT, namely the heat transport through different physical processes such as the atmospheric meridional overturning circulation (MOC), the stationary (SE) and the transient eddies (TE), also

change due to the non-CO2 paleo boundary conditions. Most notable is the increase in TE heat transport at the northern mid-latitudes, and the compensating decrease in SE heat transport. Meanwhile, over the southern mid-latitudes, a decrease in TE heat transport is identified in the data (Fig. 2b), which is not compensated by other processes, thus mainly responsible for the decrease in AHT over the Southern Hemisphere. The Southern Hemispheric AHT decrease is also linked to the aforementioned increase in OHT, consistent with the Bjerknes compensation mechanism."

8. Lines 60–63: The goals are clear but could be merged to avoid repeating ideas.

   We shortened the first goal and now it reads:

   "Our goals in this paper are: 1. To understand the early Eocene global circulation and climate. The better understanding of large-scale circulation patterns and their changes related to paleogeography helps the interpretation of proxy records."

9. Lines 75–90: Good justification for using 6-hourly data. If any spin-up was discarded, mention that explicitly.

   No, there were no spin up which was discarded. (see text under comment 10)

10. Lines 80–85: Clarify whether atmospheric initial conditions came directly from the DeepMIP runs or were re-initialized.

    Yes, the initial conditions came from the restart file of the DeepMIP simulations.

    "The atmospheric initial conditions were taken from the final state of the DeepMIP simulations, thus the simulations were direct continuations of the DeepMIP simulations in the atmosphere, which made spin up unnecessary. The ocean boundary conditions were from monthly SST as well as sea ice fraction mean values from the last 100 years of the DeepMIP simulations."

11. Lines 95–101: The tracking thresholds ( ( >1 day) 20 hPa, 1000 km distance ) are standard, but a short sentence explaining why these values are chosen would help readers not familiar with cyclone tracking.

    We include now an explaining sentence: "To focus on the significant transient eddies, our analysis includes only cyclones that persist for more than one day, travel at least 1000 km, and exhibit a pressure minimum at least 20 hPa below the global mean pressure. This filters out weak stationary pressure minima, which are often connected to orography, and ensures that we are detecting deep transient eddies which transport heat from the subtopics to the polar regions."

12.  The blocking index description is long in lines 103–112. Consider moving most of it to supplementary material and keeping a short summary here.

Thank you for the suggestion, we moved a large part of it to the Appendix and kept a shorter paragraph here.

"The hybrid blocking index evaluates the geopotential height at 500 hPa. It combines an anomaly approach with a gradient approach as described in Lohmann et al. (2024): geopotential height anomalies must exceed the climatological mean by one standard deviation to be counted as instantaneous block. To focus on persistent blocking systems, instantaneous blocks were counted as block only in the case of a minimum duration of 5 days and a minimum spatial extent of 15° in longitude and $1.5 \cdot 10^6$ km2 in space. Details of the algorithm are described in the appendix. For the climatological analysis, we calculated the mean blocking frequency defined as the percentage of days with block compared to the total number of days."

13. Line 110: Explain why you remapped to 2.5° resolution; readers may wonder whether this reduces blocking sensitivity.

The simulation of blocking in climate models is sensitive to the model resolution since blocking is a complex process and improves with increased model resolution (Schiemann et al., 2020). However, the remapping is done after running the model and does not reduce the blocking sensitivity because the blocking-like strucutre in the geopotential height field is maintained after remapping. 2.5x2.5° is a common resolution for blocking calculation (e.g. in Davini et al. (2012) where it is called "standard resolution").

We included the following in the appendix:

"The geopotential height fields were remapped to a 2.5°x2.5° grid, since this spatial resolution is the default setting of the 2D-Blocking Plugin (Richling, 2020) and a common resolution for blocking calculation (e.g. in Davini et al. (2012)). Remapping to one resolution simplifies processing and comparing several datasets with different resolution. It does not reduce the blocking sensitivity as that is connected to the original simulation resolution."

14. Lines 115–125: The explanation of NH vs SH cyclone changes is clear. Refer explicitly to Fig. 3a–d in the text to guide readers.

Thank you for the suggestion, we included the references to the corresponding figures.

15. Line 120: 36% increase and 32% decrease are large values. Clarify whether these are annual averages or seasonal means? or include uncertainty if possible.

We included in the text that these percentages are corresponding to annual averages.

"These changes are also reflected in the number of cyclones (Figure 4), with an average annual increase of 36% in the Northern Hemisphere and a decrease of 32% in the Southern Hemisphere."

16. Since you used cyclones as a stand-in for transient eddies (Line 125).. it may help to point that out more directly.

    Thank you for the suggestion, we included it now in the text:

    "The heat, transported by the mid-latitude cyclones, is what we call transient eddy heat transport in the heat transport analysis. "

17. Lines 124–130: The eastward shift in NH blocking is interesting. A short explanation of why the West Siberian Sea environment favours blocking (thermal contrast ? moisture? ) would strengthen the argument.

    We include now the following explanation:

    "The eastward shift of blocking frequency is in line with the eastward shift of the cyclone tracks to the West-Siberian Sea since the eastern boundary of cyclone tracks is a common region of blocking formation."

18. Line 129: When you say "similar intensity," specify whether this means statistically similar or just visually comparable.

    Here we only meant visually comparable, this we clarify in the text now:

    "The heat transport analysis does not show large changes in the stationary eddy heat transport at the southern mid-latitudes, which is reflected in the visually similar blocking frequencies in the early Eocene and in the pre-industrial period (Figure 5c and d)."

19. Lines 130–140: The moisture transport explanation is good. Adding a mention of storm-track–orography interaction would make it even clearer.

    We added the following sentence: "They generally produce more precipitation over oceanic regions than over land due to greater moisture availability. Moreover, enhanced precipitation also occurs when mid-latitude cyclones encounter orographic barriers, and orographic lifting takes place."

20. Increased precipitation over western North America deserves a bit more detail @Line 134. It is likely linked to orographic lifting combined with more incoming tracks.

    We added the following sentence: "Moreover, during the early Eocene the west coast of North America is experiencing high precipitations, which is probably due to a combination of the higher number of cyclone tracks in the region and the cyclones interaction with orography."

21. Lines 141–150: Nice summary of heat transport changes. You might add a simple clarifying sentence like: "These differences appear despite similar total MHT, due to Bjerknes compensation."

We changed the paragraph to: "The cyclone and blocking analysis showed that the increase of transient eddy transport over the northern mid-latitudes is due to the increase in cyclone numbers, the decrease of stationary eddy transport is due to less frequent blockings at the northern mid-latitudes. Moreover, the southern mid-latitude transient eddy heat transport decrease is connected to the decrease of cyclone numbers. This change in the Southern Hemisphere is not compensated by other processes in the atmosphere, thus effects the AHT. Nevertheless, it is compensated by the OHT through the Bjerknes compensation. Thus, the total MHT remains very similar in the pre-industrial and 1xCO2 Eocene simulations."

22. Lines 155–160: The reason for increased NH baroclinicity could be elaborated—does it come mostly from stronger thermal contrasts or increased moisture supply?

We now include the separate figure for latent and sensible heat flux (see Figure 8), and also included the following description in the text:

"Comparing the latent and sensible heat fluxes over the Eurasian region reveals that over the West Siberian Sea the two fluxes are of comparable magnitude, thus the Western Siberian Sea enhances thermal contrast and act as a moisture source, whereas over the Tethys Ocean the surface energy exchange is dominated by latent heat flux (Fig. 8)."

23. Line 160: When discussing the absence of the ACC, briefly explain the physical tie between ACC, baroclinicity, and storm-track strength.

We included the following short explanation to clarify the processes:

"Under preindustrial conditions the ACC is crucial in maintaining the strong oceanic temperature gradient between cold polar and warmer subtropical waters. This, in turn, enhances meridional temperature gradients in the lower atmosphere, which, through the thermal wind relationship, increases baroclinic instability, the fuel of mid-latitude cyclones. Under early Eocene conditions, without the ACC, the baroclinic instability decreases in the region, which is also evident by the weaker Southern Hemispheric jet stream (Figure 9 c and d). Overall, the ACC is dynamically connected to the Southern Hemispheric mid-latitude cyclones, and its absence in the 1xCO2 Eocene simulation is in line with the fewer cyclone number in the region (Figure 4)."

24. Lines 165–170: The statement that the Southern Hemisphere jet is weaker should be backed by a figure or numbers.

We included figures (Figure 9 c and d) of the 200 hPa wind filed from the pre-industrial and 1xCO2 early Eocene simulations to visualise the changes in the jet stream.

25. The link between OHT changes and deep-water formation is well made; one extra sentence tying this to energy balance constraints would be helpful (Line 170)

We rephrased this paragraph, now it reads:

"The net effect of paleogeography related changes on AHT, is a shift towards the Northern Hemisphere, which is, in turn, compensated by a southward shift in OHT. This increase in southward OHT is connected to the southern deep-water formation under early Eocene paleo boundary conditions (Zhang et al., 2022; Kelemen et al., 2023). The compensation is needed, because from the energetic perspective, a change in AHT needs to be compensated by the OHT to fulfil the top of the atmosphere energetic constraint on MHT. Thus, under the early Eocene conditions due to the AHT changes, the ocean needs to increase its heat transport over the Southern Hemisphere, which favours the development of a Southern Hemisphere-driven overturning circulation (Zhang et al., 2022)."

26. Lines 183–190: This summary is strong. A simple schematic showing the West Siberian Sea's influence on baroclinicity would make the mechanism clearer.

   We changed the sentence to:

   "In the early Eocene continental configuration, the presence of the warm and shallow epicontinental West Siberian Sea enhances air-sea interactions and acts as an extra heat and moisture source, which increases the baroclinic instability in the region."

27. Line 190: Instead of "We hypothesise…," you might soften to "These results suggest…"

   We changed the sentence: "Our results suggest that the decrease in southern AHT infers the increase in OHT, which is achieved by the Southern Hemisphere-driven overturning circulation in the ocean."

28. Line 195: The mention of future high-$CO_2$ simulations is good. Briefly state what specific questions those runs will address.

   We included now the following: "The next step of this research is the investigation of early Eocene cyclones in a high CO2 simulation, which is representing the paleo conditions more accurately. Our previous analysis (Kelemen et al., 2023) showed a slight increase in transient eddy heat transport in high CO2 simulations. Thus, we are interested to see if this is achieved through more frequent cyclones or by more heat transport per cyclone, through the warmer and moister atmosphere of the EECO climate."

29. Figures 3 and 5: These are strong figures. Adding contour lines of precipitation would sharpen this spatial feature.

   Thank you for this suggestion, we considered it (see figure below). Nevertheless, we felt, this extra information makes the plot too busy and given that Figure 6 and 7 includes similar information, we decided to keep the original Figure 3 and 5.

30. Figure 8: Latent + sensible heat fluxes are important, but splitting them or showing anomalies might make interpretation easier.

We now split the two fluxes for better interpretation (see updated figure below).

31. Figure 9: Adding wind vectors or streamlines would help readers visualize circulation, not just magnitude.

We agree, now we show the lowest model level wind speed together with wind vectors (see updated figure below).

Minor:

1. Line 128: "More dispersed distribution" Lets specify whether you mean geographically or in frequency.

We specified it: "Nevertheless, in the early Eocene the southern blockings have a more dispersed spatial distribution around Antarctica than in the pre-industrial climate."

2. Throughout....: The explanation of OHT/AHT compensation appears multiple times; consider reducing repetition!!

This explanation appears once in the introduction once in the discussion and then again at the summary. We feel it is needed in all three occasions, nevertheless we rephrased through the review the corresponding paragraphs, which we hope reduced the feeling of repetition.

3. References: Solid overall, though a couple more recent studies on Eocene storminess and SST gradients could be helpful.

We struggled to find other recent publications on Eocene storminess, and would appreciate if the Reviewer has any suggestions. Nevertheless, we now also include Agustsdottir et al. (1999) on storminess in past climates:

"It has been shown (Agustsdottir et al., 1999) that thought the geological time scale, paleogeography plays the most important role in influencing winter storm distribution."

[Figure]

Figure 3 with contour lines for annual mean precipitation.

[Figure]

Figure 8. Latent heat (a, b) and sensible heat (c, d) fluxes during boreal winter over the Northern Hemisphere in the pre-industrial (a, c) and early Eocene (1xCO2) (b, d) simulations.

[Figure]

Figure 9. Horizontal wind at lowest model level (a, b) and at 200hPa (c, d) during southern winter (JJA) over the Southern Hemisphere in the pre-industrial (a, c) and early Eocene (1xCO2) (b, d) simulations.

---

## Author Comment (AC2)

RC2: 'Comment on egusphere-2025-4923', Anonymous Referee #2, 27 Nov 2025

The authors investigated the role of paleogeography on large-scale circulation during the early Eocene. They analyzed cyclone tracks and blocking systems of the early Eocene in an atmosphere-only CESM1.2 simulation. The results show that, paleogeographic features of the early Eocene enhanced cyclonic activity at northern mid-latitudes while reduced it at southern mid-latitudes compared to modern conditions. Overall, this paper has clear structure with enough contents. I think it can be published after revision.

We thank the Reviewer for their comments and suggestions. We revised the manuscript in accordance to the Reviewers comments. Below we list our answers and the changes in the manuscript according to each comment.

Lines 3-4, Apart from these two, are there any other factors?

We agree that there are also other factors, through our analysis we found, that these two have the most influence on our subjects (cyclones and blockings). We rephased the sentence to emphasise that these are the ones we chose to focus on. "In our analysis we highlight the influence of the epicontinental West Siberian Sea as well as the impact of the lack of the Antarctic Circumpolar Current on mid-latitude cyclones and blocking events."

Lines 8-9, the details of experiments can be deleted in the abstract.

Thank you for your suggestion, we simplified this part of the abstract and deleted the not necessary details.

Line 38, 'At this time in the past', it is unclear.

We changed the sentence, now it states that we are refering to the early Eocne.
"During the early Eocene there were no currents connecting the Arctic to the Pacific basin, as the Bering Strait was closed, thus oceanic heat transport from the tropics to the Arctic flow through the West Siberian Sea (Akhmetiev et al., 2012)."

Line 65, for the first part, this paper does not separate the signals related to paleogeography and CO2, please check.

Thank you for this remark, it is right this paper does not separate signals, but focuses on paleogeography related changes, we corrected the sentence and also clarified it in the paragraph detailing our previus results in this topic.

Now the first goal sentence is:

"1. To understand the early Eocene global circulation and climate. The better understanding of large-scale circulation patterns and their changes related to paleogeography helps the interpretation of proxy records."

Line 85, why the authors use the results from the atmosphere-only model but not the coupled CESM, please state the advantages.

We note that the published coupled simulations (Zhu et al., 2019) do not have high-frequency output for cyclone and blocking event analysis. In the present study, we re-create these simulations in an efficient atmosphere-only configuration with additional 6-hourly output to track and investigate weather events.

We included the following text in the manuscript: "The atmosphere-only configuration was sufficient for our study, as we are focusing on atmospheric processes, and do not expect the ocean, which reached equilibrium in the DeepMIP simulation, to change notably in the timeline of our relatively short simulations."

Line 120, what is the relationship between cyclone distribution and the number of cyclones? For example, under early Eocene conditions, the distinction between two sets of Northern Hemispheric cyclone paths becomes less pronounced, and the number of cyclones decreases in the Northern Hemisphere.

The number of cyclones are higher under early Eocene conditions in the Northern Hemisphere, so the wider spatial ditribution of similar track density values are in line with higher cyclone numbers. We clarified it now in the text:

"However, under early Eocene conditions, the spatial distribution of cyclone tracks becomes more balanced as the distinction between the Pacific and Atlantic paths becomes less pronounced (Figure 3b). Also, there is an increase in cyclone tracks passing over land areas, especially over Europe. In contrast, in the Southern Hemisphere there are fewer cyclone tracks under early Eocene conditions compared to pre-industrial, and the track climatology shows several small density centres around Antarctica (Figure 3d), which is due to the more fragmented Southern Ocean basin. These changes are also reflected in the number of cyclones (Figure 4), with an average annual increase of 36% in the Northern Hemisphere and a decrease of 32% in the Southern Hemisphere. This makes the cyclone distribution between the hemispheres more balanced, although the average annual cyclone number is still higher in the south than in the north."

Figure 6c and d, in the northern high latitudes, the cyclone track density and high precipitation do not completely overlapping, why?

The cyclone track density only follows the cyclone center thus the whole cylone covers a larger area, and generally cyclones precipitate more over marine surface than over land, due to higher available moisture in the atmosphere.

We included now a sentence in the manuscript mentioning this: "They generally produce more precipitation over oceanic regions than over land due to greater moisture availability.

Moreover, enhanced precipitation also occurs when mid-latitude cyclones encounter orographic barriers, and orographic lifting takes place."

Line 153, how to compare the latent and sensible heat fluxes over the Eurasian region?

We now included the latent heat and sensible heat fluxes separately, which makes easier to assess their contribution, which was very similar in case of the West Siberian Sea, while the latent heat flux is more pronounced over the Tethys Ocean.

We also changed the text to:

"Comparing the latent and sensible heat fluxes over the Eurasian region reveals that over the West Siberian Sea the two fluxes are of comparable magnitude, thus the Western Siberian Sea enhances thermal contrast and act as a moisture source, whereas over the Tethys Ocean the surface energy exchange is dominated by latent heat flux (Fig. 8)."

[Figure]

Figure 8. Latent heat (a, b) and sensible heat (c, d) fluxes during boreal winter over the Northern Hemisphere in the pre-industrial (a, c) and early Eocene (1xCO2) (b, d) simulations.